# Low Cost Magnetic Field Control for Disabled People

**DOI:** 10.3390/s23021024

**Published:** 2023-01-16

**Authors:** Daniel Acosta, Bibiana Fariña, Jonay Toledo, Leopoldo Acosta Sanchez

**Affiliations:** Computer Science and Systems Department, Universidad de La Laguna, 38200 San Cristobal de La Laguna, Spain

**Keywords:** robotics, intelligent wheelchair, handicapped

## Abstract

Our research presents a cost-effective navigation system for electric wheelchairs that utilizes the tongue as a human–machine interface (HMI) for disabled individuals. The user controls the movement of the wheelchair by wearing a small neodymium magnet on their tongue, which is held in place by a suction pad. The system uses low-cost electronics and sensors, including two electronic compasses, to detect the position of the magnet in the mouth. One compass estimates the magnet’s position while the other is used as a reference to compensate for static magnetic fields. A microcontroller processes the data using a computational algorithm that takes the mathematical formulations of the magnetic fields as input in real time. The system has been tested using real data to control an electric wheelchair, and it has been shown that a trained user can effectively use tongue movements as an interface for the wheelchair or a computer.

## 1. Introduction

According to Faria [1], an intelligent wheelchair is a mobility device that helps people with disabilities. It is equipped with an intelligent control system that allows the user to control its movement. Some key characteristics of an intelligent wheelchair include:The interaction with the user is made using some devices such as joysticks, voice, computer vision, and other sensors.The wheelchair is equipped with a set of sensors and algorithms that allow it to navigate autonomously in various environments.The wheelchair has the ability to communicate with other devices such as doors, other intelligent wheelchairs, and electronic devices such as televisions.

There are currently numerous prototypes of intelligent wheelchairs being developed, most of these projects, as described by Faria [1] often focus on either improving the navigation of the prototype or enhancing the ease of use of the user–machine interface.

One of the main challenges in designing an intelligent wheelchair is developing an effective human–machine interface (HMI) [2]. For users who have some hand movement, a joystick attached to the wheelchair is a straightforward and easy-to-use HMI that provides good speed, ergonomics, and autonomy. However, for users with severe disabilities, other sensors may be necessary. These can include blow sensors or mouth joysticks.

For individuals with severe disabilities, traditional interfaces may not be appropriate. An experimental technology that is currently receiving a lot of attention is a brain–computer interface (BCI). BCIs are often based on electroencephalography (EEG), which measures brain activity using electrodes [3]. While this method is still experimental and may be too complex for some users, more practical and simple user interfaces are being developed as an alternative.

As previously stated, our research presents a solution that employs a magnet attached to the tongue of the user as a non-invasive control method for an intelligent wheelchair. The magnet is secured to the tongue using a suction pad or glue. With proper training, the user is expected to be able to use the movements of their tongue to mimic a joystick which allows the user to control the wheelchair or other devices connected to the control computer. A low-cost magnetometer is used to measure the magnetic field and determine the position of the magnet within the user’s mouth. The configuration, shown in Figure 1, is ergonomic and easy to use for the user, with the magnet positioned on the tongue and the sensor placed close to the mouth like a headset microphone.

Magnetic fields have been utilized as a method for indoor localization in various studies, such as [4,5], which use the unique magnetic fields present in an area as “fingerprints” for local localization. A survey of different methods for magnetic indoor localization with a focus on fingerprinting can be found in [6].

In [7], short-range magnet localization is tested and evaluated for accuracy, and a simulation tool is developed to estimate the configuration needed for a desired level of accuracy. The paper presents the theoretical concepts and is designed for magnetic fields generated by coils, rather than permanent magnets. It is not specific to any particular application but rather presents a general localization system.

Several similar prototypes have been described in the literature, such as in [8], where a set of Hall Effect sensors are placed around the tongue using a teeth protector. The position of the magnet attached to the tongue is obtained using these sensors. However, this device has a few problems: the tongue position is not continuously obtained and the sensors are often large, difficult to install, and uncomfortable. In [9], an array of eight three-axis anisotropic magnetoresistive sensors are placed in front of the user’s mouth, and the position of the magnet is measured using this array. The prototype is large and uncomfortable, making it impractical for daily use. The position of the magnet in the mouth is calculated using a Levenberg–Marquardt minimization algorithm based on data from all the sensors. In [10], external magnetic interference needs to be attenuated to reduce the effect of external magnetic fields on the magnetic sensor, requiring a high-computational processor using FPGA to obtain the position based on the measurements. In [11], the performance of an intraoral inductive sensor is tested. The system is based on 18 inductive sensors, and the study demonstrates the effectiveness of this type of sensor for people with severe disabilities. In [12], RFID tags in the form of tongue proximity sensors are placed over the palate to sense tongue position. This is used to facilitate tongue control of different devices, and user training rate and ability are measured. In [13], a similar approach is used, but two magnetometers close to the mouth are required to sense the position of the magnet. This makes the prototype more complex and uncomfortable. The magnet position is calculated using the field difference between two magnetometer measurements. The calculations are performed on a computer outside the system, which limits its applicability.

The novelty of this paper is to design and test a human–machine interface based on only one sensor close to the user’s mouth. Other prototypes described in the bibliography use many sensors close to the mouth, which makes it a big and uncomfortable device for the user. The use of only one magnetometer simplifies the placement of the device and improves its ergonomics. The magnet position inside the mouth is calculated on a low-cost microcontroller, so no extra hardware, such as a computer or specific electronic, is necessary. Another novelty is the use of a second magnetometer to measure ambient magnetic fields and remove them from the mouth magnet, this improves the accuracy and usability in different environments.

This paper is organized as follows: First, the physical concepts of the magnetic field and how to measure it are presented. The device, sensor, hardware, and configuration are presented including the algorithms used to calculate magnet position. Some results are presented measuring the accuracy of the system and the conclusions are drawn.

## 2. Magnet Field

The magnet used in the tongue localization can be approximated by a magnetic dipole model [14,15]. We can assume that the magnet is composed of two charges, +b and −b, with a distance of *d*, so the module of the magnetic field $\vec{\mu }$ is $\left|\mu \right|=bd$. These parameters can be calculated experimentally and it depends on the magnet characteristics. Figure 2 shows the magnetic field generated by the magnetic dipole bd in two dimensions (x,z) in the position $\vec{p}$. It is composed of the contribution of charges +b and −b modulated by the distance *r*.

### 2.1. Magnetic Field in z Axis $B_{z}$

Equation (Equation 1) shows the field generated by a magnet in the function of the distance *r* with both charges aligned to the *z* axis, assuming a separation between the charges of *d*.
(1)$$B_{z}=B_{z\left(b+\right)}+\;B_{z\left(b-\right)}=\frac{b}{r_{+}^{2}}\frac{z-\frac{d}{2}}{r_{+}}-\frac{b}{r_{-}^{2}}\frac{z+\frac{d}{2}}{r_{-}}=b\left(\frac{z-\frac{d}{2}}{r_{+}^{3}}-\frac{z+\frac{d}{2}}{r_{-}^{3}}\right)$$

Assuming that *d* is much smaller than *r*, we can obtain an expression of the distance $1/r^{3}$ as shown in Equation (Equation 2) that can be used to calculate the magnetic field received at a distance *r*. As we can see, the magnetic field decreases in cubic order with distance $1/r^{3}$, so this kind of sensor can only be used for very short ranges, as in the current application.
(2)$$\begin{array}{c} \frac{1}{r_{+}^{3}}\approx \frac{1}{r^{3}}\left(1+\frac{3zd}{2r^{2}}\right)\\ \frac{1}{r_{-}^{3}}\approx \frac{1}{r^{3}}\left(1-\frac{3zd}{2r^{2}}\right) \end{array}$$

Combining Equations (Equation 1) and (Equation 2) we obtain an approximate value for the magnetic field value as shown in Equation (Equation 3). Applying $\left|\mu \right|=bd$ and $\cos\left(\theta \right)\;=\frac{z}{r}$ we obtain the final field equation, which depends on the distance and angle to the point $\vec{p}$.
(3)$$B_{z}\approx \frac{bd}{r^{3}}\left(3\frac{z^{2}}{r^{2}}-1\right)=\frac{3\left|\mu \right|}{r^{3}}\left({cos}^{2}\left(\theta \right)\;-\frac{1}{3}\right)$$

### 2.2. Magnetic Field in x Axis $B_{x}$


$B_{x}$ is composed of the influence in the *x* axis of both charges in the point $\vec{p}$ as shown in Figure 2.
(4)$$B_{x}=B_{x\left(b+\right)}+\;B_{x\left(b-\right)}=\frac{b}{r_{+}^{2}}\frac{x}{r_{+}}-\frac{b}{r_{-}^{2}}\frac{x}{r_{-}}=bx\left(\frac{1}{r_{+}^{3}}-\frac{1}{r_{-}^{3}}\right)$$

Combining Equations (Equation 2) and (Equation 4) and using the definition of sin() and cos() function for the θ angle, we obtain the magnetic field expression for axis *x*
$B_{x}$ as shown Equation (Equation 5)
(5)$$B_{x}=3\frac{bl}{r^{3}}\frac{xz}{r^{2}}=\frac{3\left|\mu \right|}{r^{3}}sin\left(\theta \right)cos\left(\theta \right)$$

### 2.3. Magnetic Field in a Point in the Space

In a general case, where the magnet is not aligned to the *Z* axis, the magnetic field associated with the rotation angles according to *Z* is shown in Equation (Equation 6), where φ is the azimuth angle in spherical coordinates, and it represents the orientation angle between the magnet and the *z* axis.
(6)$$\begin{array}{c} B_{x}=\frac{k}{r^{3}}\left(sin\theta cos\left(\theta \right)cos\left(\varphi \right)\right)\\ B_{y}=\frac{k}{r^{3}}(sin\left(\theta \right)cos\left(\theta \right)sin\left(\varphi \right)\\ B_{z}=\frac{k}{r^{3}}\left(cos^{2}\left(\theta \right)-\frac{1}{3}\right) \end{array}$$

The magnetic field $\vec{B}$ in a point *p* can be characterized in spherical coordinates as shown in Equation (Equation 6). The paper’s objective is to localize a magnet placed over the user’s tongue. A magnetic sensor is used to measure 3D magnetic field coordinates. The angles (θ,φ) and the distance *r* characterize the position of the magnet with respect to the coordinate system placed on the magnetic sensor.

In Equation (Equation 6), $k=3\left|\mu \right|\;$ is the constant of the magnet. *k* can be obtained by measuring $\vec{B}$ in positions where $\theta =90^{\circ }$, for the positions $B_{x}=0$, $B_{y}=0$ and $\vec{B}=B_{z}=\frac{k}{3r^{3}}$. Figure 3 shows a graphical plot of the magnet placed at different distances from the sensor. The *x* axis shows the distance as $\frac{1}{r^{3}}$ and the *y* axis the magnetic field value at that distance. Using these data, the magnetic constant *k* is measured for each magnet. The idea is to characterize each individual magnet for each specific sensor to obtain the magnet properties measured by each sensor and compensate for the possible sensor variances. This procedure is conducted for each sensor and for each magnet used by this sensor, obtaining an accurate measurement of the magnet for each unit of the sensor. The variances in sensor measurements due to the tolerances in the fabrication process are removed in this way and the final accuracy is improved.

#### 2.3.1. Obtaining θ from a Magnetic Measurement

In this project, all the localization is made using a low-cost microcontroller, so we need a mathematical expression to solve the magnet position. Numeric solvers are not allowed due to the low computation power of our device. The final objective is to obtain an analytic expression to obtain magnet position based on magnet dipole properties.

The objective is to localize the magnet based on its magnetic field. If we use the magnetic field equations shown in Equation (Equation 6), we can obtain the following equations to calculate the θ angle.
(7)$$\begin{array}{c} \sqrt{B_{x}^{2}+B_{y}^{2}}=\frac{k}{r^{3}}\;sin\left(\theta \right)cos\left(\theta \right)\\ q=\frac{\sqrt{B_{x}^{2}+B_{y}^{2}}}{B_{z}}\\ q=\frac{\sin\left(\theta \right)\;\cos\left(\theta \right)\;}{{cos}^{2}\left(\theta \right)\;-\frac{1}{3}} \end{array}$$

The variable *q*, as defined in Equation (Equation 7), is used to obtain θ from the magnetic field measured $\left(B_{x},B_{y},B_{z}\right)$. *q* presents a discontinuity if the angle θ has a value of $54.73^{\circ }$, so this must be analyzed during the calculus. Table 1 shows the behavior of the *q* denominator based on the angle in the proximity of the discontinuity.

Table 1 shows that if $abs({cos}^{2}\left(\theta \right)-\frac{1}{3})<0.02$ the angle θ will be $55^{\circ }$. The main problem is that we do not know the actual value of $abs({cos}^{2}\left(\theta \right)-\frac{1}{3})$, but we know $B_{z}$ from the magnetic sensor, so we can estimate it by multiplying by $\frac{r^{3}}{k}$. So, the worst case for *p* is calculated to obtain a condition for $B_{z}$ to avoid a zero division. If we suppose that *r* is small, for example, *r* = 2 cm, the value of $\frac{k}{r^{3}}$ will be high. With a *k* of approximately 90, $\frac{k}{r^{3}}$ will be 11.25, this value will generate a $abs(B_{z})<0.2$ and θ will be $55^{\circ }$ obtaining a zero division.

Assuming that θ is different from $55^{\circ }$ we continue to obtain the value. For that, a variable change is made according to Equation (Equation 8).
(8)$$m=cos\left(\theta \right);sin\left(\theta \right)=\sqrt{1-m^{2}}$$

So, *q* can be redefined as Equation (Equation 9):(9)$$q=\frac{m\sqrt{1-m^{2}}}{m^{2}-\frac{1}{3}}$$

Operating this equation and making the variable change $n=m^{2}$ a second-degree function is obtained with two valid solutions.
(10)$$\left(q^{2}+1\right)n^{2}-\left(\frac{2q^{2}}{3}+1\right)n+\frac{q^{2}}{9}=0$$

To obtain the correct value for θ we can use the $B_{z}$ sign as Equation (Equation 11) shows. Only one θ will accomplish Equation (Equation 11), so this θ will be selected.
(11)$$sign\left(B_{z}\right)=sign\left(cos^{2}\left(\theta \right)-\frac{1}{3}\right)$$

#### 2.3.2. Obtaining *r* and φ

Knowing θ, we can directly obtain the value of *r* from Equation (Equation 6) as Equation (Equation 12) shows.
(12)$$r=\sqrt[{3}]{k\frac{cos^{2}\left(\theta \right)-\frac{1}{3}}{B_{z}}}$$

The angle φ can be obtained directly from Equation (Equation 6) knowing the relation between $B_{x}$ and $B_{y}$ as Equation (Equation 13) shows.
(13)$$\varphi =arctan(\frac{B_{y}}{B_{x}})$$

Using this procedure, the magnet location coordinates (θ,φ,r) can be obtained directly. Knowing the position of the magnet, coupled with the user’s tongue, it can be used as a human–machine interface.

## 3. Hardware Setup

In this section, we will describe the components used in the prototype. The measurement process is controlled using an Arduino microcontroller to obtain all the data from the magnetic sensor. In this case, we have used an Arduino UNO with an ATmega328 microcontroller, which is a low-cost option that meets all the requirements of this project.

The proposed device consists of the following main parts:2 HMCL5883L Magnetometer with an estimated price of USD 2.Arduino Uno based on AtMega328 with an estimated price of USD 5.MC14053 analog multiplexer with an estimated price of USD 1.

So the price of the whole system, including the support for the magnetometers, the magnet, and other components such as resistors is less than USD 15. The design is made by selecting the parts to obtain a low-cost device with enough accuracy.

### 3.1. HMC5883L Sensor

The main part of the system is an HMC5883L magnetic sensor. This sensor was originally designed as an electronic compass for mobile devices. It is able to measure the ambient magnetic field in 3D (X,Y,Z). It is based on a high-sensibility anisotropic magneto-resistive sensor and it includes a module for the automatic demagnetize of the sensor. The magnetic field is captured by a 12-bit analog-to-digital converter. The circuit is controlled by an I2C bus for configuring and reading data.

The intensity of the magnetic field decays in $\frac{1}{r^{3}}$ way, as Figure 4 shows, so the accuracy of the system depends on the distance between the sensor and the magnet. The HMC5883L sensor has an in-circuit programmable gain, so the lecture of the sensor can be adjusted to obtain a good resolution. If the magnet is close to the sensor, the gain is reduced avoiding saturating the sensor. If the magnet is far, the gain is augmented to obtain a good resolution. The maximum resolution with the maximum gain is 0.73 miliGauss/bit, and the used magnet has a magnetic field of 10,000 Gauss.

Figure 4 represents the field measured in magnetometer reads including the in-circuit adjustable gain. These measures have been standardized by the gain in which they were captured. The values close to the sensor are obtained using a small gain, so the real measurement is the sensor read divided by the gain. The measures away from the sensors are captured with a high gain, so the value of the reading is reduced. In this way Figure 4 represents the range of the measurements. The resolution of the samples depends on the distance to the sensor, but in this case, at a distance of 10 cm, the measure is 1300 and the error is less than 1 mm in localization. The whole working area is measured with enough resolution for the proposed application.

The HMC5883L is sampled to its maximum frequency, 160 Hz, and every measure is applied to a mean filter, obtaining an output filtering frequency of 7.5 Hz. This frequency is enough to sense the movements of the tongue and it works like a low-pass filter to integrate variable magnetic fields that could be present in the environment. The system was tested in the presence of continuous and variable magnetic fields and the two-magnetometer schema was able to remove background fields and obtain an accurate localization.

The final design includes two sensors placed in different positions and with different orientations. The first one, far from the magnet, is called the reference magnetometer M1, the second one is called the measurement magnetometer, M2 is placed close to the user’s mouth. The M1 magnetometer measures the ambient magnetic field including the earth’s magnetic field and all the other magnetic noise present in the environment. The M2 magnetometer measures the magnet and the environment field. In order to remove the residual magnetic field and obtain only the magnet field, magnetometer M1 is used to remove the environment field from the M2 measure. The position and orientation between the sensors can change when the user uses the device, so a calibration process is performed before measurements.

As Figure 4 shows, the decay of the magnetic field with distance is $\frac{1}{r^{3}}$, so the distance between both magnetometers to avoid the reference magnetometer measuring the magnet inside the mouth is 15 cm. At that distance, the magnet field is barely measured by the magnetometer. In our prototype, both magnetometers are 20 cm apart which ensures the independence of the measurements.

In the final version of the prototype, an MC14053B analog multiplexer is used to communicate between both sensors. The I2C address of the sensor cannot be changed, so two sensors cannot be connected to the same bus. The analog multiplexer switches the clock and data lines from the bus to M1 first and then changes the lines to connect to the M2. A digital output pin is used to change the multiplexer lines. In this way, both sensors with the same address can be read in the same bus without collisions.

### 3.2. Measurement Device

The prototype is shown in Figure 5. It includes the analog multiplexer, two HMC5883L magnetometers, I2C pull-up resistors, and connection cables. The software in the microcontroller measures M1 and M2 alternatively using the multiplexer and sends the data to a controller PC using the USB bus. The design of the prototype is similar to a headset microphone, with magnetometer M1 placed in the back of the neck and M2 close to the user’s mouth. This design is comfortable and easy to install.

Using the two magnetometer schema, the prototype is robust against the influence of the external magnetic fields. The prototype has been tested intensely indoors and outdoors and the effect of the external magnetic field is neutralized in all the circumstances. For example, our laboratory has a lot of electrical equipment and iron furniture, and these do not affect the accuracy of the measurements. The case of huge magnetic fields is very rare, and they reduce their intensity as $\frac{1}{r^{3}}$ with distance, so at a few meters of distance, this field can be canceled using the measurement magnetometer. In the worst case, the device would not work properly near a huge magnetic field, but this situation is very rare and it has not been tested by the authors.

The localization algorithm presented in Section 2.3.1 and Section 2.3.2 is designed to be implemented in a microcontroller with a small computation power, in this case, inside an Arduino Uno. There is a mathematical expression for solving each of the magnet coordinates, so no numeric or optimization methods are necessary. This gives us the ability to obtain real-time results from low-computation power devices. The algorithm error is negligible and the largest source of error comes from the sensor measurements and it is less than 1 mm at 10 cm of distance.

The sensor HMC5883L can be configured to measure with different gains. The code is designed to dynamically adjust the gain to obtain the maximum accuracy without saturation. In this way, if the lower measurement of the three channels is less than 1500, in the 12-bit analog-to-digital converter, the gain is incremented until the measurements fit in the channel. On the other hand, if the measurements are saturated, with a lecture of 4096 in any of the channels, the gain is reduced until obtaining a non-saturated value. Using this strategy, the measurement always has good accuracy and saturation is avoided.

### 3.3. Measurement Conversion from Reference Magnetometer to Measurement Magnetometer

The use of two magnetometers in our system allows us to measure the environment field and the magnet field independently. The two magnetometers are placed close, in the sense that the same environment field will be captured by both; however, the field of the magnet is not received by the M1 magnetometer because it is placed far enough from the magnet and $\frac{1}{r^{3}}$ magnet field reduction with distance reduce the effect in a short distance. The measurement of the M2 magnetometer includes the environment field, and sometimes this field can be higher than the magnet field. The M1 gives us the measurement of the environment so we can remove it from the measurement of the M2 which includes magnet and environment. The main problem is that the rotation matrix between the magnetometers should be obtained. Using this matrix, the measurement of the M1 magnetometer can be rotated to the M2 coordinates and the removal of the environment field can be performed directly in M2 coordinates. The calibration process between the two magnetometers is based on [16].

The first step is to take the N measurement of each magnetometer. These measurements are $P_{1}\left(i,j\right)$ for M1 and $P_{2}\left(i,j\right)$ for M2 with i=1:3 and j=1:N. The objective of the algorithm is to find the rotation matrix *R* between these two sets of measurements, so we can convert the measurement taken from the M1 coordinate system to M2.

The centroids of the measurements are defined in Equation (Equation 14).
(14)$$C_{1}\left(i\right)=\frac{1}{N}\sum_{j=1}^{N}P_{1}\left(i,j\right);C_{2}\left(i\right)=\frac{1}{N}\sum_{j=1}^{N}P_{2}\left(i,j\right);i=1,2,3$$

We create the *H* matrix based on the measurements taken from the magnetometers as Equation (Equation 15) shows.
(15)$$H=\sum_{j=1}^{N}\sum_{i=1}^{3}\left(P_{1}\left(i,j\right)-C_{1}\left(i\right)\right)\left(P_{2}\left(i,j\right)-C_{2}\left(i\right)\right)$$

To obtain the rotation matrix, a singular value decomposition (SVD) is calculated, converting the *H* matrix as the product $H=VSU^{T}$. *V* and *U* are orthogonal matrices and *S* is a diagonal matrix composed of the singular values. The rotational matrix *R* can be calculated as $R=UV^{T}$.

One of the main challenges is that the singular value decomposition (SVD) algorithm is programmed on an Arduino microcontroller, which limits the computational resources available. Therefore, a stable and low-computational cost algorithm is needed to calculate the SVD of the 3 × 3 dimension rotation matrix *H*. The algorithm to obtain the SVD of the *H* matrix is as follows:Calculate the eigenvalues and eigenvectors of the symmetric matrices $B=H^{T}H$ and $B^{T}=HH^{T}$.The eigenvectros of $H^{T}H$ compose the columns of the matrix *U* while eigenvectors of matrix $HH^{T}$ compose columns of matrix *V*.Singular values of *S* are the roots squares of the eigenvalues of $H^{T}H$.

To implement this procedure, the calculation of the eigenvalues and eigenvectors of matrix *B* must be performed. The initial approach was to use the Gram–Schmidt decomposition of matrix QR, as this implementation is relatively simple and has a low computational cost. However, this method showed unstable numerical behavior even with 3x3 matrices. As a more robust alternative, the QR decomposition based on Givens rotations was used.

Givens rotations can be considered to be numerically stable. With this method we introduce zeros into the matrices in order to obtain a triangular matrix, in this case, the matrix $B=[b_{ij}]$. To transform *B* into an upper triangular matrix three multiplications using Givens rotation matrices are performed. The first step uses the $G_{1}$ rotation matrix to zero out the third row and the first column, as shown in Equation (Equation 16). This process is repeated for the other elements in the matrix until it is fully upper triangular.
(16)$$G_{1}=\left[\begin{array}{ccc} c & 0 & -s\\ 0 & 1 & 0\\ s & 0 & c \end{array}\right];A=G_{1}B;r=\sqrt{b_{11}^{2}+b_{31}^{2}};c=\frac{b_{11}}{r};s=-\frac{b_{31}}{r}$$

The second step gives a zero in the second row and the first column by multiplying by $G_{2}$ according to Equation (Equation 17).
(17)$$G_{2}=\left[\begin{array}{ccc} c & -s & 0\\ s & c & 0\\ 0 & 0 & 1 \end{array}\right];D=G_{2}A;r=\sqrt{a_{11}^{2}+a_{21}^{2}};c=\frac{a_{11}}{r};s=-\frac{a_{21}}{r}$$

The third step gives a zero in the third row and the second column by multiplying by $G_{3}$ according to Equation (Equation 18).
(18)$$G_{3}=\left[\begin{array}{ccc} 1 & 0 & 0\\ 0 & c & -s\\ 0 & s & c \end{array}\right];r=\sqrt{d_{22}^{2}+d_{32}^{2}};c=\frac{d_{22}}{r};s=-\frac{d_{32}}{r}$$

Matrices *R* and *Q* are defined in Equations (Equation 19) and (Equation 20).
(19)$$\begin{array}{c} R=G_{3}D\\ Q=G_{1}^{T}G_{2}^{T}G_{3}^{T}\\ B=QR \end{array}$$
where
(20)$$\begin{array}{c} B=QR \end{array}$$

After many tests, with n=20 iterations, the accuracy of the rotational matrix is fixed below 1 $\times 10^{-9}$. This precision is far beyond what is necessary. The results are compared to a standard implementation in Octave, given similar results for each tested case. The implementation based on Gram–Schmidt gives, in some cases, very large errors.

### 3.4. Communication Module

The Arduino microcontroller is connected to python software installed in the robotic prototype to communicate with the device. The software is designed to receive information on magnet localization from the Arduino microcontroller. This communication is based on ad hoc messages defined in our communication protocol over Arduino serial links. The software allows us to change some parameters using a GUI (Graphical User Interface) and it implements an ROS (Robotic Operative System) module in order to control a robotic prototype using the sensor.

The main functions of the Arduino control library are:setCenter: Sets the reference system for the control system.readPos: Receive from Arduino the current pose of the magnet and send it as a ROS messsage.setRotation: The calibration between the two magnetometers is made in the Arduino software. After matrix *R* is obtained, the Arduino starts to use to calculate the magnet field and remove the environment field.

### 3.5. ROS Connection

Robot Operating System (ROS or ros) is an open-source robotics middleware suite. Although ROS is not an operating system (OS) but a set of software frameworks for robot software development, it provides services designed for a heterogeneous computer cluster such as hardware abstraction, low-level device control, implementation of commonly used functionality, message-passing between processes, and package management. It is the way that our intelligent wheelchair is implemented. The ROS module receives information from the Arduino system, and sends it to the system in a standard message as Figure 6 shows, the intelligent wheelchair receives the message and it makes the decision to transport the user to its destination.

The objective of this paper, an intelligent wheelchair, is shown in Figure 7. The wheelchair is composed of a localization system based on encoder odometry and other sensors [17]. An algorithm based on BIM technology to indicate the destination of the intelligent wheelchair [18] and a system for obstacle avoidance and localization [19].

The wheelchair is based on the following sensors:Odometry. Wheel odometry uses incremental encoders coupled to the motors. The system captures the movement of the wheels in real time. It is one of the main sensors for a robotic prototype and the basis for any localization system [20]. The position and orientation are obtained by integrating wheel displacement.IMU. An IMU uses a set of sensors to estimate inertial magnitudes. In this case, an electronic gyroscope, which measures angular speed, is integrated to estimate orientation. The IMU also has an electronic compass to determine the north direction, but the behavior of this sensor can be erratic indoors due to magnetic interference, and it is not used in the intelligent wheelchair.Lidar. LIDAR is used to detect obstacles and to build a map for navigation and localization.

The set of sensors used in the wheelchair is shown in Figure 7. Circled in orange are the lidars, a pair of Sick TiM 551, valid for indoor and outdoor use, with a maximum range of 8 m, an angular resolution of 1 degree, a field of view of 270 degrees, and a measurement frequency of 15 Hz. In green is the IMU unit, located under the seat. It is an MPU9250 which includes an accelerometer, gyroscope, and magnetometer in a single chip. In red is the position of the wheels where the encoders are coupled to calculate the odometric movement.

The wheelchair is programmed using the ROS system. The schematic of the modules is shown in Figure 8. It receives the desired destination from the user, and the wheelchair makes the decision to transport the user in a secure way to its destination. It is composed of a localization module, a SLAM (simultaneous localization and mapping module) or a pre-build map-based navigation module, and an obstacle avoidance and path planning module which generates the commands to apply to the wheelchair. The commands are received in this case from the magnet placed in the user’s tongue. This is converted to an ROS message that the rest of the system processes. The command indicates the movement direction, and all the systems apply this command in an intelligent way, avoiding obstacles or making decisions to achieve the final destination.

The software reads the position of the magnet and publishes the position as a ROS message. It works similarly to a joystick, where tongue position is the command generator. If the magnet is placed in the center of the mouth, no command is sent, if it moves forward, reverse, right, or left, the module sends the corresponding message according to Figure 9. The movement of the tongue is translated to a command message, and the conversion is linear. So if the user moves the tongue a little to the left, the wheelchair will send a command to move a little to the left, but if the user moves a bigger distance of the tongue to the left, he will be indicating to the wheelchair a bigger displacement, so fine movement control can be achieved. The information is sent as a message and the rest of the intelligent wheelchair system receives the command. The intelligent wheelchair will follow this command in an intelligent way, avoiding obstacles, making decisions, and selecting the best path to the destination.

## 4. Results

In this section, some tests with the sensor module are presented. The tests include the calculus of the rotation matrix, the measurement of the environment magnetic field, the measurement of the magnet field, and the calculation of the magnet position according to the algorithm presented in previous sections.

The magnet is placed in different positions and the localization measured by the system is obtained. In Table 2, the values of magnet positions in Cartesian coordinates (x,y,z) are shown, the same position in spherical coordinates (r,θφ) and the position obtained by the localization algorithm using the magnetometers, $\left(r_{m},\theta_{m},\varphi_{m}\right)$ are presented. As we can see, the system works well until a distance of 6cm, which is enough distance to measure all the points inside the mouth. If we increase magnet power, using a bigger magnet, we can increase the measured distance. The reference magnetometer is about 15 cm from the magnet, so the influence of the magnet field in this magnetometer is not appreciated.

Table 3 shows the error obtained between the actual measurement and the value obtained by our sensor. The formula used to obtain the error is shown in Equation (Equation 21) where the relative error value expresses as % depends on the value of the results.
(21)$$e_{x}=100\frac{abs(x-x_{m})}{x}$$

Table 3 shows that the error in the distance is quite good, proportionally bigger on short distances due to the application of Equation (Equation 21) and the division by the measurement. With respect to θ errors, they are quite small at short distances, but they increase with distance. In the working area, the error is bounded, so the prototype can be used with accuracy. The worst behavior is with the φ angle, but this magnitude is less important for the control of devices and the error is bounded to less than 15%.

In Figure 7, the final test with the real prototype is shown. The prototype shown in this paper was tested in a real environment with an intelligent wheelchair developed by our research group. The basic commands forward, right, left, and reverse were tested, and the velocity of the command was modulated by the position of the tongue in the mouth. The computer shows the position of the tongue as user feedback in the GUI generated for this application.

The user can drive the wheelchair in a real environment, traveling through complicated areas such as doors and corridors with a little training before starting. As results show, the behavior is very good in the working area, making the sensor suitable as a human–machine interface for people with disabilities.

## 5. Conclusions

In this paper, a short-range localization system based on a magnet is presented. The magnet is attached to the tongue of the user, and they can use it to control any device, in this case, it is used as a control system for an intelligent wheelchair.

The magnetic field is measured using a magnetometer. Another magnetometer, placed about 15 cm away from the magnet, is used, for the measurement of the environment magnetic field. The system can measure the magnet about 5 cm from the sensor, enough range from tongue movements measurements.

Using the magnetic field of the magnet, the position is obtained, and the magnet is located in the mouth.

All the calculus is made inside an Arduino microcontroller, which makes our sensor independent of a computer, so it can be used to control any device. The limitation of computational power forces us to obtain a mathematical expression for the position of the magnet based on the measurement of the magnet field. In this way, a real-time magnet position can be calculated in the microcontroller, so there is no need for specific hardware or software.

The results show the accuracy of the method, measuring the magnet position, in real tests. The device is small, discrete, ergonomic, and easy to use. It can be used easily by any user.

The device was tested successfully to move an intelligent wheelchair designed for the movement of people with disabilities. The system is based on ROS, and the user uses it as a joystick to send commands to the actual wheelchair.

## Figures and Tables

**Figure 1 sensors-23-01024-f001:**
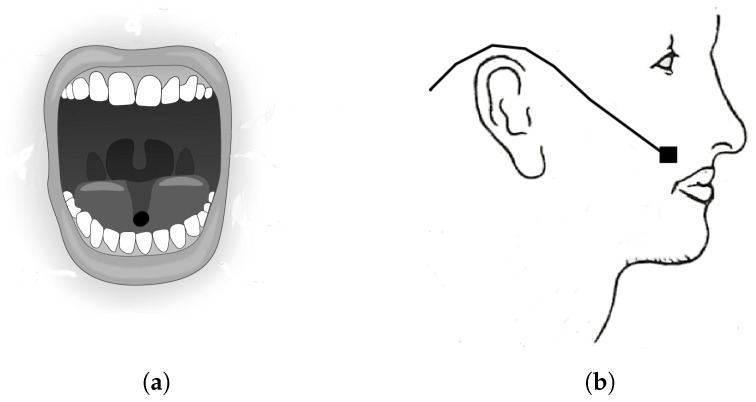
Prototype configuration: (**a**) Position of the magnet in the tongue; (**b**) Magnetic sensor close to the mouth.

**Figure 2 sensors-23-01024-f002:**
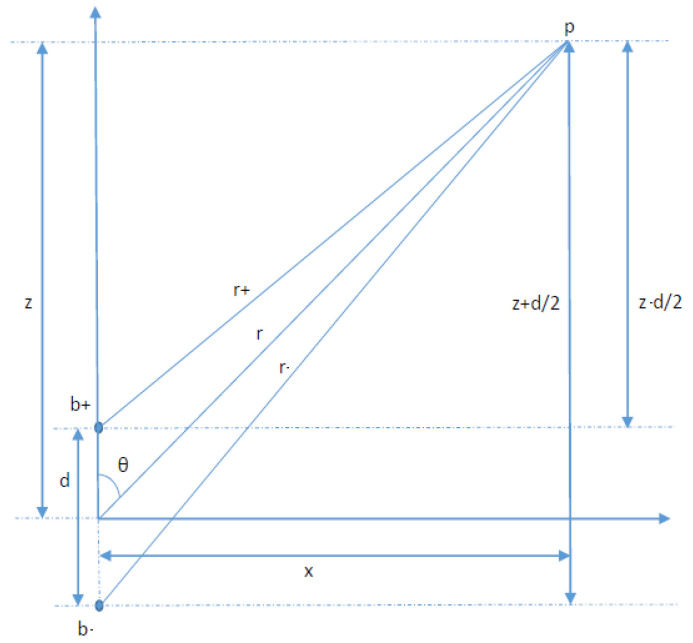
Magnetic field of a magnetic dipole.

**Figure 3 sensors-23-01024-f003:**
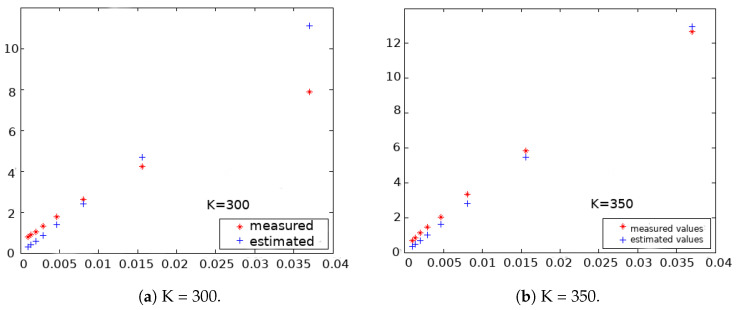
Magnetic constant calculation for different magnets.

**Figure 4 sensors-23-01024-f004:**
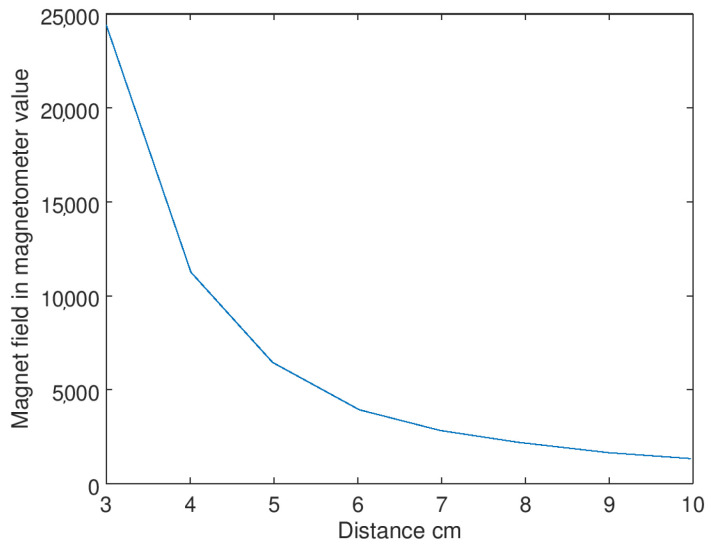
Magnetic field in the function of distance.

**Figure 5 sensors-23-01024-f005:**
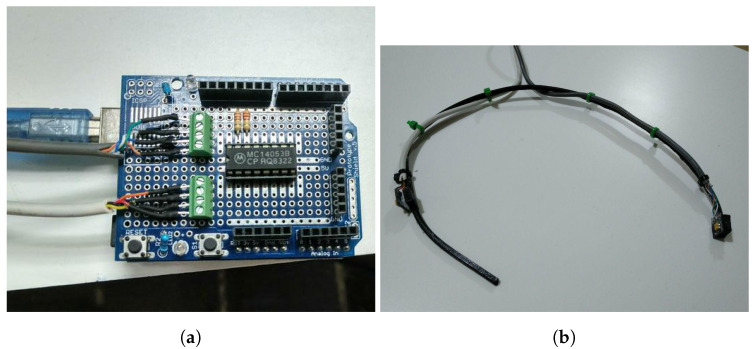
Final prototype configuration: (**a**) Connections between magnetometers and microcontroller; (**b**) Sensors M1 and M2 in the final configuration as a headset microphone.

**Figure 6 sensors-23-01024-f006:**
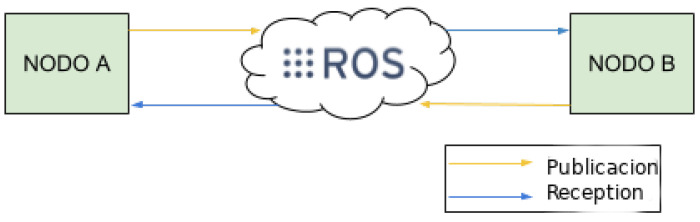
ROS system schema.

**Figure 7 sensors-23-01024-f007:**
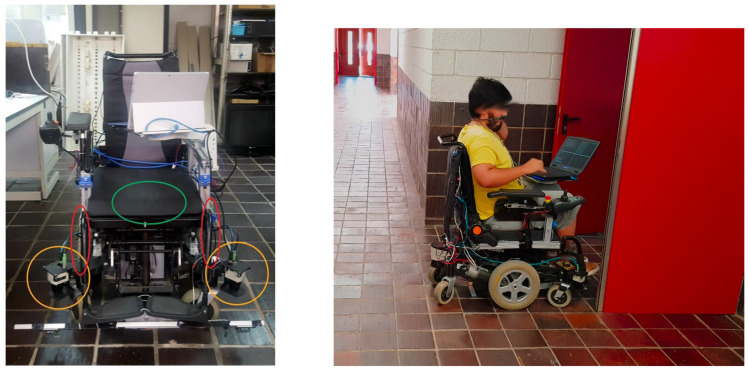
Intelligent wheelchair controller by the magnetic device.

**Figure 8 sensors-23-01024-f008:**
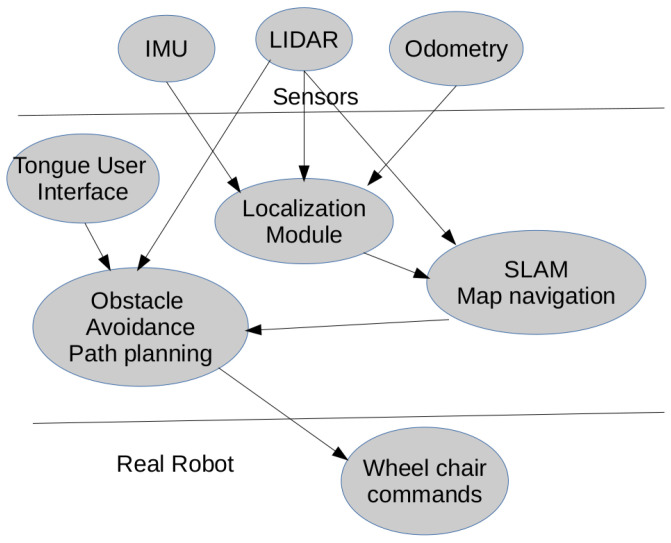
Intelligent wheelchair working schematic.

**Figure 9 sensors-23-01024-f009:**
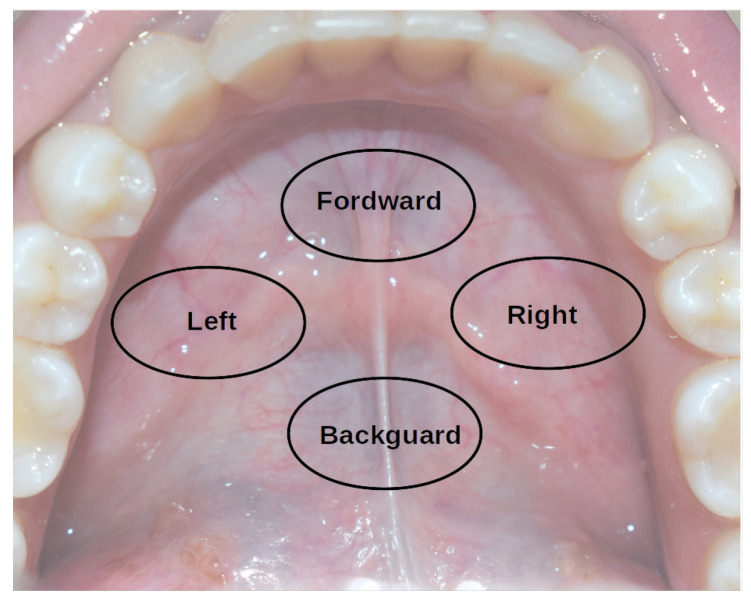
Conversion schema from tongue position to command.

**Table 1 sensors-23-01024-t001:** *q* values in the function of θ.

θ (Degrees)	*q* Denominator
52	0.0457
53	0.0288
54	0.0122
55	−0.0043
56	−0.0206
57	−0.0367

**Table 2 sensors-23-01024-t002:** Actual measurements with the magnet localization system.

x (cm)	y (cm)	z (cm)	r (cm)	r${}_{m}$ (cm)	θ (o)	θ${}_{m}$ (o)	ϕ (o)	ϕ${}_{m}$ (o)
0	3.0	1.5	3.3	3.6	63.4	62.4	90.0	76.8
0	4.0	1.5	4.3	4.3	69.4	69.3	90.0	74.5
0	5.0	1.5	5.2	5.1	73.3	74.5	90.0	70.0
0	6.0	1.5	6.2	5.6	76.0	76.8	90.0	61.3
0	3.0	3.0	4.2	4.1	45.0	48.1	90.0	83.1
0	4.0	3.0	5.0	5.1	53.1	56.7	90.0	83.7
0	5.0	3.0	5.8	6.2	59.0	54.4	90.0	87.7
3	0	3.0	4.2	4.1	45.0	49.1	0	13.2
4	0	3.0	5.0	5.0	53.1	55.0	0	8.0
5	0	3.0	5.8	5.8	59.0	52.7	0	0.0

**Table 3 sensors-23-01024-t003:** Error table between actual results and magnet measurements.

r (cm)	r${}_{m}$ (cm)	e${}_{r}$ (%)	θ (o)	θ${}_{m}$ (o)	e${}_{\theta }$ (%)	ϕ (o)	ϕ${}_{m}$ (o)	e${}_{\phi }$ (%)
3.3	3.6	9.1	63.4	62.4	1.6	90.0	76.8	14.7
4.3	4.3	0	69.4	69.3	0.2	90.0	74.5	17.2
5.2	5.1	2.0	73.3	74.5	1.6	90.0	70.0	22.2
6.2	5.6	9.7	76.0	76.8	1.1	90.0	61.3	31.9
4.2	4.1	2.4	45.0	48.1	6.9	90.0	83.1	7.7
5.0	5.1	2.0	53.1	56.7	6.8	90.0	83.7	7.0
5.8	6.2	6.9	59.0	54.4	7.8	90.0	87.7	2.6
4.2	4.1	2.4	45.0	49.1	8.9	0	13.2	13.2 *
5.0	5.0	0	53.1	55.0	3.6	0	8.0	8.0 *
5.8	5.8	0	59.0	52.7	10.7	0	0.0	1.0 *

* values correspond to the abs of the error.

## Data Availability

Not applicable.

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
