# Peer review of "Low Cost Magnetic Field Control for Disabled People"

_sensors, 2023, doi:10.3390/s23021024_

Round 1

Reviewer 1 Report

The article presents research on a system for tracking the position of a magnet in a wheelchair control application. The work has a logical structure and contains all the necessary elements, including: an introduction to the research area along with the presentation of the research background, theoretical analysis of the considered issue, the measurement system along with the analysis of its work, control tests, final conclusions. The work is interesting, but some elements are worth considering before making the final decision. Below are comments for the authors to consider in the next version of the article.

1. In the introduction, I miss clearly highlighting the novelty of the proposed work in relation to other works on the topic? What does this way of positioning bring in the context of the state of the art?

2. It is worth reviewing the editing part of the work, improving the graphic presentation of figures and language and syntax errors in the text.

3. Figures 6 and 7 can in principle be combined. It is worth adding close-ups of the key elements of the installation of such a wheelchair. In the case of a system description, it is worth adding a block diagram of the entire system showing its components. Similarly, it is worth considering the scheme of operation of the developed algorithm.

4. At the beginning of the work, the authors notoriously characterize their system with the term low-cost. This term in no way defines the engineering properties of the system. Such a description is in the further part of the work, so please either refer to the rest of the work in the aforementioned fragments of the initial text, or quote what is meant by the term low-cost.

5. How does the bit resolution and the sampling frequency of the microcontroller affect the precision of magnet position calculation? Did you analyse the error range?

6. What do you mean by „localization algorithm is optimized”? In what sense? What is the precision of the calculation you have considered and what error level it may bring to the assessment of the position?

7. What is the required distance for the second sensor to not sense the field of the magnet? What level is set to be the threshold value?

8. In a situation where the magnetic field of the environment is many times greater than the magnetic field, which results in the need to set low sensitivity in both field sensors to sense the field without saturation, how accurate is the measurement of the field from the magnets? Isn't this field too low, and acts as low-level background to the environmental field in this case?

Reviewer 2 Report

The paper aroused my curiosity, although many questions remained unanswered. I list a few:

1) the authors claim that the second magnetic sensor is used to subtract environmental magnetic interference from the measurement of the first sensor, which has to measure the field produced by the magnet. However, the sensors applied can measure only very low frequency or static fields. Among the static ones, I guess the earth's magnetic field is not relevant compared to that of the magnet on the tongue.

2) a Multiplexer is used to manage the two sensors on the same I2C bus, having the same built-in address. Justified, although magnetic sensors with programmable address could be searched for.

3) The meaningful of the sentence (row 272) "The intensity of the message is correlated by the tongue displacement, so a fine control can be made by the user" is missing.

4) The description of the wheelchair control system is missing. The sentence "The Arduino microcontroller is connected to a python software installed in the robotic prototype to communicate with the device" is not demostrated.

5) The movements of the tongue should control the direction of wheelchair's movements, not the final destination. All this is misunderstanding. The sentence "The ROS module receive information from Arduino system, and sends to the system in a standard message as figure 5 shows, the Intelligent wheelchair receives the message and it makes the decision to transport the user to its destination." is not clear, how can it?

6) The orientation of the magnet on the tongue, from what I understand, should not indicate the direction of path, but rather forward/backward/left/right, therefore a measurement mapping system would be necessary.

Round 2

Reviewer 2 Report

The authors have fully replied to my observations, the paper in the present form can be accepted.